# Ultrastructural and Immunohistochemical Characterization of Maternal Myofibroblasts in the Bovine Placenta around Parturition

**DOI:** 10.3390/vetsci10010044

**Published:** 2023-01-07

**Authors:** Valentina Kuczwara, Gerhard Schuler, Christiane Pfarrer, Louiza Tiedje, Ali Kazemian, Miguel Tavares Pereira, Mariusz P. Kowalewski, Karl Klisch

**Affiliations:** 1Institute of Veterinary Anatomy, Vetsuisse Faculty, University of Zurich, 8057 Zurich, Switzerland; 2Clinic for Veterinary Obstetrics, Gynecology and Andrology of Large and Small Animals, Faculty of Veterinary Medicine, Justus Liebig University Giessen, 35392 Giessen, Germany; 3Institute for Anatomy, University of Veterinary Medicine Hannover, Foundation, 30559 Hannover, Germany

**Keywords:** cow (*Bos taurus*), gestation, term, interstitium, myofibroblast, contraction, alpha-smooth muscle actin, angiotensin-II

## Abstract

**Simple Summary:**

The placenta is a constantly changing organ that faces continuous tissue remodeling throughout gestation, reflected by recurrent stromal adaptation. The myofibroblasts, contractile stromal cells, are key players involved in orchestrating physiological tissue remodeling. Nevertheless, details regarding ultrastructural and functional aspects of myofibroblasts in the developed bovine placental stroma remain veiled. Here, the presence of myofibroblasts in the maternal placental stroma was confirmed by immunohistochemical and ultrastructural analysis, with the co-localization of several factors, e.g., *α* -smooth muscle actin, vimentin and nuclear progesterone receptor. During short-term tissue culture, the contraction of placentomal caruncle tissue slices was induced by angiotensin-II. Furthermore, a three-dimensional reconstruction demonstrates an example of a bovine placental myofibroblast. According to the present findings, it is postulated that the maternal stroma of the placenta in the cow contains a high proportion of contractile myofibroblastic cells. Herewith, a basis for further investigations regarding the functional aspects and the role of myofibroblasts in peri- and post-partum placental events in cattle was created.

**Abstract:**

Myofibroblasts are contractile cells that exhibit features of both fibroblasts and smooth muscle cells. In the synepitheliochorial placenta of the cow myofibroblasts are found in the maternal stroma. However, a deeper understanding of the structure and function of the stromal myofibroblasts in the developed bovine placenta is still missing. Thus, immunohistochemical and ultrastructural analyses in bovine term placentomes, compared to non-pregnant caruncle samples, were conducted. To investigate functional aspects, contractility of placentomal caruncle slices was assessed in an in vitro contraction assay. Additionally, a three-dimensional reconstruction of a bovine placental myofibroblast was created. Immunofluorescent staining revealed a characteristic pattern, including cytoplasmic expression of *α*-smooth muscle actin, strong perinuclear signal for the intermediate filament vimentin and nuclear progesterone receptor staining. Ultrastructurally, stress fibers, extended cisternae of the rough endoplasmic reticulum and perinuclear intermediate filaments were observed. Moreover, in vitro stimulation with angiotensin-II, but not with prostaglandin F2*α*, induced contraction of placental caruncle tissue. Altogether, these results indicate that progesterone-responsive myofibroblasts represent a mesenchymal phenotype that is involved in the contractile properties of bovine placental stroma. Therefore, the present findings suggest a potential involvement of myofibroblasts in post-partum events of cattle, i.e., expulsion of fetal membranes and uterine involution.

## 1. Introduction

Myofibroblasts are fibroblastic cells showing features of both fibroblasts and smooth muscle cells. This cell type was firstly described to play a crucial role in formation of granulation tissue and wound healing and considered to be responsible for wound contraction and scar formation in response to mechanical forces in several tissues [1,2,3]. Fibroblastic cells react by acquiring specialized contractile features, associated with connective-tissue remodeling, and differentiate into myofibroblasts [4,5]. Initially, the myofibroblast induction was known as a stromal reaction against pathological conditions, also in the uterine tissue of several species [6]. However, several studies showed that myofibroblast-like cells also appear in non-pathological uterine and placental tissue of healthy individuals of various species [6,7,8].

The function and ultrastructural components of myofibroblasts are still under debate. Recently, the overall agreement on stress fiber formation as the single most important criterion was stated to identify activated myofibroblasts [9]. This definition seems reasonable, since myofibroblasts generally contain a contractile machinery, which indicates the importance in wound closure of healing acute wounds, as well as time-limited and spatially restricted scarring of the tissue [10]. The force needed for wound contraction is generated by an assembly of myofibroblasts, which together form a contractile structure. Thus, it is well known that mechanical tension is crucial for myofibroblast modulation and maintenance of their contractile activity [10,11]. Increased formation of stress fibers, more precisely the de novo expression of alpha-smooth-muscle actin (*α*-SMA) filaments, enables the development of contractile activity [12]. In addition to that, the occurrence of contractile elements and the strong positive signal for *α*-SMA indicates the close relation between myofibroblasts and smooth muscle cells [13]. Myofibroblasts may have heterogeneous origins, however, their development follows a well-established sequence of events [14]. Myofibroblast differentiation from fibroblasts and contraction is induced by mechanical stress, accompanied by stress fiber formation, and a conductor, e.g., the transforming growth factor-beta-1 (TGF-β-1) [10,15]. The effect of TGF-β depends on the presence of fibronectin (FN), in detail, its splice variant ED-A FN in the extracellular matrix (ECM) [12,14,16]. So far, a previous study on TGF-β expression in bovine term placentomes showed, that TGF-β1 receptor (TGFB1) is characteristically localized in the maternal septa of the caruncle [17]. A deeper understanding for the underlying transition process of myofibroblasts in the bovine placenta stroma is still missing.

Surrounded by ECM in the stromal tissue, myofibroblasts are closely allied to each other, communicating via different protein complexes at their contact-sites. To investigate the intercellular interaction, it was demonstrated that intercellular mechanical coupling of stress fibers via adherens junctions improves contraction of myofibroblasts in collagen gels. Furthermore, besides mechanical adherens junctions, fibroblastic cells possess the ability to communicate electrochemically via gap junctions [18]. In these regards, formation of gap junctions was already demonstrated at the ultrastructural level between myofibroblasts of wound granulation tissue [18,19]. Among localized intercellular connections, gap junctions are the ones that allow the exchange of ions and small molecules between myofibroblasts and fibroblasts, cardiomyocytes, mast cells, and, possibly, the endothelium [11]. Besides Connexin 45, these connecting gap junctions are primarily formed by Connexin 43 (Cx43) [20,21,22,23,24].

Another functional aspect of myofibroblasts is their contractability. As a component of the renin-angiotensin systems (RAS) [25,26], Angiotensin (Ang) II (Ang-II) stimulates fibroblast migration and contraction through, e.g., Ang-II receptor type 1 (AT1), as previously demonstrated in human granulation tissue [27]. Several studies have also provided evidence for the presence of all components of local tissue RAS in the uteroplacental unit in humans as well as in other species [25,26,28,29]. In this respect, a previously published study demonstrated high densities of the two main Ang-II receptor types, AT1 and AT2 receptors, in all parts of the bovine placenta and fetal membranes [25]. In detail, AT2 receptors were highly expressed in the fetal part of the placenta, whereas AT1 receptors predominated in the maternal part, suggesting that Ang-II exerts an effect on regulatory as well as growth processes in these tissues [25]. Furthermore, prostaglandins (PGs) are central mediators involved in several female reproductive functions, establishment and maintenance of pregnancy, as well as parturition in most mammals [30,31,32]. Prostaglandin F2 alpha (PGF2*α*) is known as a luteolytic hormone and myometrial stimulant [30]. Results of previous studies suggested that changes in the expression of PG relaxant or contractile receptors could be involved in the maintenance of uterine quiescence for most of the gestation and activate the uterus to contract at the time of parturition for expulsion of the fetus [32]. In these regards, investigated temporal and tissue-specific expression of PGF2*α* receptors at the maternal-fetal interface suggested a selective and distinctive role for PGF2*α* in uterine activities during pregnancy in bovine [32].

The bovine placenta consists macroscopically of about 100 discrete mushroom-shaped sites, the placentomes, in which intense contact between fetal villi (cotyledon) and maternal crypts (caruncle) occurs. Specialized fetal binucleated trophoblast cells fuse with uterine epithelial cells to form trinucleated cells in the uterine epithelium. This fusion led to the classification of the bovine (ruminant) placenta as synepitheliochorial [33]. The feto-maternal contact zones are exposed to great mechanical forces during bovine pregnancy [8]. Therefore, the establishment of a strong supportive system within the placentomal tissue, especially in the maternal aspect, is necessary to provide mechanical stability. In the bovine placenta, characteristic alterations in ECM composition and distribution, including collagen remodeling, have been reported to occur during the various reproductive stages [34,35]. In this respect, similar to fibroblasts, myofibroblasts can produce and modify their surrounding ECM, including fibronectin (FN). Hence, they participate in a complex, interactive dialogue between themselves and their microenvironment, as corroborated previously [36]. Most knowledge about the structure and cellular differentiation of placental myofibroblasts comes from research on humans and rodents. In the human and murine placenta, e.g., myofibroblasts are of mesenchymal origin, confirmed by the expression of the mesenchymal cell marker vimentin (VIM) [37,38], which further suggests their relation to fibroblasts and decidual cells. Accordingly, a previous study demonstrated that human placenta derived decidual stromal cells also show expression of VIM, *α*-SMA as well as representative ultrastructural similarities with myofibroblasts [7]. Similar to nuclear progesterone (P4) receptor (PGR) expressing decidual cells, which are described in invasive placentae, like in humans, rodents and dogs [39,40,41,42,43], there is also evidence of nuclear PGR expression in *α*-SMA positive cells in the caruncular stroma of bovine placenta throughout gestation and around parturition [44,45]. Altogether, it can be stated, that fibroblasts provide origin for different cells, including myofibroblasts and decidual cells in different species and placental types.

Despite the great importance of cows in the dairy and meat production industry, there is sparse information available on the cells of the maternal stroma in the bovine placenta, or rather the placentomes [34]. In this context it would be important to gain a better understanding of the role of the maternal stroma throughout the separating and involuting processes post-partum [46,47]. Although, previous studies investigated caruncular stromal cells of fetal bovine uteri [48], little is known about the fully developed, non-pathological bovine placenta around parturition. Therefore, the aim of this study was to elucidate morphological and ultrastructural features of stromal myofibroblasts in the bovine endometrial caruncle stroma near term. In addition to that, obtained ultrastructural findings were visualized in a representative three-dimensional (3D) reconstruction of the maternal stroma myofibroblast, using serial block-face scanning electron microscopy (SBF-SEM, [49]).

## 2. Materials and Methods

### 2.1. Tissue Sample Collection

Samples from cows (*Bos taurus*) were used in this study. Detailed information about collection, reference, number of animals and grouping are listed in Table 1.

### 2.2. Immunofluorescence (IF) Staining and Confocal Microscopy

Normal term placentome samples, formalin fixed, paraffin-embedded and grouped in pregnant (P) prepartal and parturient individuals (each n = 3), were used for IF staining and subsequent confocal microscopy. The material was part of a previous study [50]. Cows in the prepartum observation group underwent cesarean section (273–282 gestation days (gd)) immediately after the prepartum decline in maternal P4 became evident, as determined by regular measurements at eight-hour intervals during the last two weeks of gestation. No signs of active labor were observed at the time of surgery. Placentomes from parturient cows were obtained from cows that underwent routine cesarean section for fetal pelvic malposition after spontaneous onset of labor. In these animals, placentomes were collected immediately after calf removal, which in all cases were normally developed and vital. Fixation and preparation procedure for the paraffin blocks has been published previously [50]. Caruncle tissue of non-pregnant (NP) individuals in different developmental stages was collected at the local slaughterhouse. These samples were allocated to three groups (each n = 3), i.e., NP post-partum, NP nulliparous, NP multiparous. For histological analysis and IF staining, harvested samples were fixed overnight in 4% neutral buffered formalin (Formafix, Switzerland) and washed with 1 × PBS for one week followed by dehydration in a graded ethanol series. The tissue was processed with a Leica TP1020 spin processor (Leica Biosystems Switzerland AG, Muttenz, Switzerland) and embedded in paraffin. Three µm sections were cut and mounted onto Superfrost TM Plus slides (Gerhard Menzel, Braunschweig, Germany).

Formalin fixed paraffin embedded sections of all tissue samples were utilized for triple-colour IF staining, using first steps of a previously published immunohistochemistry procedure [51] followed by standard IF protocol [44,52]. Briefly, tissue was deparaffinized, rehydrated and washed under tap water for 5 min. Heat-induced epitope retrieval was performed using 10 mM citrate buffer (pH 6.0). Slides in retrieval buffer were heated in a microwave oven at 600 W for 15 min followed by 20 min cooling at ambient temperature. After blocking of nonspecific binding with 10% normal goat serum (Cat. -Nr.: 5560–0007, LGC Seracare, Milford, MA, USA), slides were incubated with primary antibodies, diluted in IHC buffer with 0.3% Triton X, pH 7.2–7.4 (0.8 mM Na2HPO4, 1.47 mM KH2PO4, 2.68 mM KCl, 137 mM NaCl.) overnight at 4 °C in Shandon’s Coverplates. IHC buffer-washed slides were subsequently incubated with fluorochrome-conjugated secondary antibodies at ambient temperature for 1 h. Detailed information regarding the used antibodies is provided in Table 2. Nuclei were visualized by adding 4′,6-diamidino-2-phenylindole (DAPI; Sigma-Aldrich Chemie GmbH, Buchs, Switzerland) to the secondary antibody. Consequently, slides were washed and rinsed using 4% formalin and distilled water. Glass cover slips were finally mounted with Glycergel (Dako North America, Inc., Carpinteria, CA, USA). Negative controls were prepared according to previously published examples [43,53,54], i.e., staining with only primary or secondary antibodies, as well as omitting any antibody, to check for autofluorescence. Additionally, isotype controls, a combination of either mouse IgG or IgG2a antibody and rabbit IgG antibody were applied. Sections were qualitatively assessed and documented with a confocal laser scanning microscope (SP8, Leica Microsystems GmbH, 35578 Wetzlar, Germany).

### 2.3. Transmission Electron Microscopy (TEM)

For TEM, bovine placental specimens taken at parturition (n = 3), with normal release of fetal membranes, were utilized. The material was part of an earlier study [34]. Likewise, to the collected caruncle tissues (3 NP groups, each n = 3) that were used for IF staining, additional samples (NP grouping, each n = 3) were employed for TEM procedures. Preservation and fixation procedures followed standard protocols, described previously [44,55].

Fixation of the collected NP tissue samples was performed as published previously [55]. In short, samples were perfused through uterine blood vessels with fixative (2.5% glutaraldehyde and 4% paraformaldehyde in 0.1 M of Sorensen’s phosphate buffer), cut into slices of 1 mm thickness and further fixed for 2 h. Samples were post-fixed in osmium tetroxide and embedded in Epon. An ultramicrotome (Ultracut E, Reichert, Vienna, Austria) was used to cut 1 μm semithin sections, which were stained with toluidine blue. The processed sections were used to identify well-preserved areas in the tissue. Resin blocks of selected regions of interest (ROI) were trimmed and cut to ultrathin sections (70 nm) with the ultramicrotome. Mounted on grids, sections were stained with lead citrate and uranyl acetate. Sections were observed with a transmission electron microscope (CM12, Philips, Eindhoven, Holland) and pictures were taken with a CCD camera (Orius SC1000, Gatan, Pleasanton, CA, USA).

### 2.4. Serial Block-Face Scanning Electron Microscopy (SBF-SEM) and 3D Reconstruction

For SBF-SEM an image stack of a bovine placenta sample was used. The sample was collected from a placentome of a pregnant Holstein Friesian cow (gd 278 days), terminated by cesarean section, which was performed after rupture of fetal membranes. Preservation in Karnovsky’s fixative was followed by embedding in Durcupan’s resin, as previously described [56]. These were recorded with a section thickness of 60 nm. SBF-SEM images were used for further processing in 3D reconstruction and modeling of the target cell type.

ROIs with well-preserved areas of maternal stroma were selected on single SEM surface scans of the block. The selected image field, showing the target cells, was utilized for the creation of an image stack with 970 images. 3D reconstruction was performed on a defined stack of 180 images (Voxel size = 15 × 15 × 60 nm), which included a complete myofibroblast. Herewith, a manual segmentation of the nucleus, the stress fibers and the cytoplasm, was conducted, using the ImageJ software (ImageJ 1.53q, bundled with 64-bit Java, version 1.8.0_172) plugin TrakEM2 [57]. For visualization, segmented images were merged and resulting channels were used to create a 3D model (Imaris ×64, version 9.9.1, Bitplane AG).

### 2.5. Tissue Contraction Assay (CA) and Histology

For the contraction experiments, placentomes (n = 3) were collected at a local abattoir. Pregnancy stages were estimated based on the fetal crown-rump length [58] and ranged between late second and early third trimester (gd 156, 174 and 220). Placentomes were manually separated to obtain caruncle tissues, that were transported in cold PBS (4 °C) within 30 min to the laboratory for further processing.

Caruncles were rinsed with cold PBS and kept in DMEM high glucose medium (Gibco, Thermo Fisher Scientific AG, Reinach, Switzerland), supplemented with 3.7 g/L NaHCO_3_ during all processing steps. Tissue slices were then obtained using a Krumdieck Tissue Slicer MD6000 (TSE-Systems GmbH, Bad Homburg, Germany) vibratome, following the manufacturer’s recommendations. In short, cylindrical tissue cores (∅ = 8 mm) of caruncles were obtained using a motorized tissue coring press MD5000 (TSE-Systems GmbH, Bad Homburg, Germany). The tissue core was then directly transferred to the cylindrical holder of the vibratome, and tissue slices (300 μm) were cut and transferred into 12-well-plates (one slice per well) (TPP Techno Plastic Products AG, Trasadingen, Switzerland) for further experiments.

Tissue slices were initially incubated for 30 min in serum-free DMEM High glucose medium (Bio Concept, Allschwil, Switzerland) supplemented with 100 U/mL penicillin and 100 μg/mL streptomycin (PAN Biotech, Aidenbach, Germany), 1% insulin-transferrin-selenium (ITS; Corning from Thermo Fisher Scientific AG, Reinach, Switzerland) and 0.01% bovine serum albumin (BSA; SUB001, Canvax Biotech, Córdoba, Spain), under standard culture conditions (37 °C, 5% CO_2_ in air, in a humidified incubator). After the preincubation period, and prior to stimulation (time 0), photographs of the incubated slices were taken with a digital microscope VHX-6000 (Keyence Deutschland GmbH, Neu-Isenburg, Germany). Subsequently, culture medium was replaced with medium containing no adds (control), or containing either 0.25 μM, 0.5 μM or 1 μM of PGF2_*α*_ (38562-01-5, Santa Cruz Biotechnology, Inc., Dallas, TX, USA) or Ang-II (05-23-0101, Sigma-Aldrich Chemie GmbH, Buchs, Switzerland). All experiments were run in duplicate, i.e., two slices were stimulated for each experimental condition. Digital pictures of the treated slices were taken after 1 h, 2 h, 4 h and 6 h of stimulation. Tissue areas (mm^2^) were measured with VHX-6000 software (2016 Keyence Corporation, Ver. 3.2.0.121, System Ver. 2.07). To normalize possible user variation, each slice was measured twice, and average values were used. Results are presented as percentage (% of ∆1 h–∆6 h) of tissue slices’ area variation (shrinkage).

Additionally, to evaluate the morphology and possible structural changes in tissue constitution induced by treatment, slices were washed at end of treatment (6 h after stimulation) with ice-cold PBS and fixed in 4% phosphate-buffered formaldehyde for 10 min, being then washed with PBS overnight at 4 °C before tissue processing. Control slices (collected at time 0) were also fixed for histological analysis. Tissue slices were then processed and paraffin-embedded using a standard protocol and stained with hematoxylin and eosin. Stained slides were then visually analyzed, and representative pictures were taken, using Leica DMRXE light microscope equipped with a Leica Flexacam C1 camera (Leica Microsystems, Wetzlar, Germany).

### 2.6. Statistical Analysis

GraphPad 3.06 Software (GraphPad Software, San Diego, CA, USA) was employed for all the statistical analyses. Parametric one-way ANOVA was applied to evaluate significant differences in time-course experiments of in vitro contraction stimulation of caruncle. When *p* was less than 0.05, ANOVA was followed by the Tukey-Kramer multiple comparisons post-test. Numerical data are presented as arithmetic means +/− standard deviation.

## 3. Results

### 3.1. Immunohistochemical Characteristics of Myofibroblasts in the Bovine Placenta Stroma around Term and in Caruncle Tissue of Non-Pregnant Cows

#### 3.1.1. *α* -SMA and VIM Expression Pattern

Myofibroblasts, located in maternal septa, showed a distinct immunostaining with the antibody against *α* -SMA, with no dissimilarities in the examined tissue samples and stages (Figure 1). Immunoreactivity was detected intracellularly in the cell periphery, following the course of either the long processes of the cell or forming compacted fiber bundles. The cells were accompanied by some stromal vessels, showing the tunica media, with circular oriented *α*-SMA positive smooth muscle cells. In addition to that, apparently less positive staining signals were also found in the vessels of fetal placental compartments. In the two different sample groups (P and NP), varying double-staining patterns could be observed (Figure 1, left panel).

VIM showed a similar expression pattern in all tissue sections within the examined stages and groups (Figure 1). Briefly, cytoplasm of maternal stromal cells was positive for VIM. All sections presented a perinuclear staining pattern for VIM (Figure 1). Therefore, it appeared, that VIM filaments were mainly present in the perinuclear cytoplasm of the stromal myofibroblasts, located around the nuclei and peripherally surrounded by *α*-SMA positive filaments. Fetal components (stroma and endothel) in placentomes were negative for VIM, except positive signal in fetal vessels (Figure 1, left and right panel). Additionally, there are apparent differences in staining density of VIM in the NP caruncle tissue sections compared to the placentomal tissue samples (Figure 1, left and right panel). Generally, an apparently increased immunolabeling intensity could be detected in all cell types of the caruncle stroma, not specifically in myofibroblasts (Figure 1, last row).

#### 3.1.2. Progesterone Receptor (PGR) Expression Pattern

Myofibroblasts in bovine placenta and caruncle showed similar positive nuclear PGR staining (Figure 1, right panel). This immunostaining pattern was only detected in a subpopulation of VIM-positive cells and apparently presented increased density within the NP caruncular stroma compared to the maternal placenta stroma. No signal for PGR was detected in fetal cells of the placental sections.

#### 3.1.3. Connexin 43 Expression Pattern

When double-stained with nuclear PGR, positive focal (spot-like) signals for the gap junctional marker Cx43 were observed in myofibroblasts of the maternal placenta stroma. In detail, cytoplasmic periphery and loci along the cell membrane borders showed binding of the Cx43-antibody (Appendix A). Strong positive signals of high density were present amongst the PGR-positive subpopulations in the non-pregnant caruncle sections, whereas signs of Cx43 expression appeared weaker in the pregnant tissue samples around term.

#### 3.1.4. Fibronectin Expression Pattern

Similar to the Cx43 stained sections, stromal cells appeared in a highly organized connective meshwork when stained with nuclear PGR, surrounded by moderate cytoplasmic FN signal in the triple stained NP caruncle sections (Appendix A). In contrast, expression of co-localized markers differed in density and intensity in the maternal stroma of the pregnant placentome tissue. In the placentome PGR-positive cells presented weaker staining signals for FN in their cytoplasm compared to the non-pregnant caruncle sections.

### 3.2. Ultrastructural Characterization of Myofibroblasts in the Bovine Maternal Placenta Stroma around Term and in Caruncle Tissue of Non-Pregnant Cows

#### 3.2.1. Representative Features of Myofibroblasts

In the TEM sections, we identified myofibroblasts in the maternal stroma of preserved pregnant or non-pregnant tissue sections (Figure 2). These cells showed an elliptical shaped nucleus, surrounded by branched cytoplasm with several processes of different length Furthermore, three characteristic features of myofibroblasts could frequently be observed in both placentomal and caruncular tissue samples. First, cytoplasmic microfilaments formed stress fibers, which were associated with either dense bodies or with focal adhesions at the cell membrane (Figure 2B,D–F). Second, the cells showed expanded cisternae of endoplasmatic reticulum (ER; Figure 2A–C,E,F). Third, intermediate filaments surrounding the nucleus of the cell were frequently detected, showing a perinuclear localization (Figure 2C, inset). Several samples of different pregnant individuals around parturition and collected non-pregnant sample sections showed no differences in ultrastructural characteristics.

#### 3.2.2. Ultrastructural Network and Grouping of Myofibroblasts in Caruncle Tissue

Especially in the non-pregnant caruncle, stromal cells demonstrated epitheloid character and formed an organized network. Similar to fibroblasts, they were directly coupled to each another, demonstrated by ultrastructural features of gap junctions in adjacent myofibroblasts (Appendix A).

#### 3.2.3. D Reconstruction of Placental Myofibroblast

One typical myofibroblast was chosen within the selected ROI of the SBF-SEM stack. Representative features, including expanded ER cisternae, stress fibers and perinuclear intermediate filaments were observed. Following 3D reconstruction (Figure 3), the myofibroblast presented as an elongated cell, showing various processes and stress fibers predominantly in the cell periphery. These processes or extentions were varying in lenghts and sizes. Stress fibers could frequently be detected as either strands or bundels, which were most commonly seen close to the cytoplasmic membrane, revealing a reticular construction in the whole cell model.

### 3.3. Ang-II Stimulates Contraction in Placental Caruncular Tissue Sections In Vitro

Caruncle slices incubated with Ang II (Figure 4), but not those with PGF2_*α*_ (*p* > 0.05, Appendix A), showed a significant (*p* < 0.05) or highly significant (*p* < 0.01) time dependent size reduction in all tested concentrations (Figure 4). In detail, shortly after starting Ang-II incubation, moderate but not significant tissue size reduction was present after 1 h or 2 h (∆1 h or ∆2 h; Figure 4A,B). Hence, the slice shrinkage increased greatly after 4 h of stimulation, represented in significant size reduction of 0.25 μM Ang-II treated slices, compared to control (*p* < 0.05, Figure 4C). The strongest reaction, associated with severe tissue contraction, was seen in the Ang-II treatment group after ∆6 h of incubation. In these slices a very significant shrinkage was detected at all tested treatment concentrations of 0.25 μM, 0.5 μM and 1 μM Ang-II (*p* < 0.05 for each concentration; Figure 4D), in comparison to control. No significant area size reduction was observed in the control group, that was run simultanously to the Ang-II treatment group (*p* > 0.05, Figure 4; Appendix A).

Additional evaluation of the cell morphology and general tissue constitution was performed, using H&E-staining of formalin fixed paraffin embedded slices, either before or after contraction stimulation by Ang-II and PGF2_*α*_. The tissue showed only rare signs of cell death or apoptosis (Appendix A). There was no indication for hypoxic damage in all tissue slices of each treatment group. Low contamination of the caruncle by fetal tissue was observed. Compared to the control slices (0 h, 6 h), there was no evident difference in tissue composition after the 6 h stimulation process. Since no remarkable contraction effect was observed in the PGF2_*α*_ treated slices, H&E-staining results are not presented. Nevertheless, morphological assessment revealed similar patterns to Ang-II treated slices.

## 4. Discussion

Our study focuses on immunofluorescent and ultrastructural hallmarks of myofibroblasts in the bovine placenta. Given their contractile potential, myofibroblasts appear to be interesting target cells associated with wound healing processes in various tissues. After detachment of the fetal membranes post-partum, the caruncles might be regarded as big “wounds”. This is associated with activation of cell populations necessary for tissue remodeling and wound healing. To date, there is still limited knowledge on the specific expression pattern and co-localization of myofibroblast-markers in bovine uterine tissue. Previous studies, performed on maternal stromal cells in caruncles of developing, prenatal uteri [48] or on bovine term placenta tissue [44], highlight the presence of *α*-SMA-, VIM- and PGR-expression in the bovine placental stroma. Accordingly, the purpose of this study was to provide deeper insights into a myofibroblast’s ultrastructure in the maternal stroma during advanced pregnancy and around parturition in comparison with NP caruncle samples.

*α*-SMA appears to be a useful marker for myofibroblasts also in maternal stromal cells in non-pregnant bovine endometrial caruncles [6]. Overall, by using three-color IF in the present study, caruncular stromal cells could be recognized as myofibroblasts in both P and NP groups. Maternal stromal cells of the developed bovine placentome showed positive cytoplasmic double staining of *α*-SMA filaments and mesenchymal cell marker VIM. Intermediate filaments represent one of the most resilient and strongest natural structures [59,60]. As presented previously, intermediate filaments, including VIM filaments, are key players for tissue integrity. Given the fact that they form a highly developed meshwork of adjustable filaments they react to mechanical stimuli in challenged tissues and induce changes in cell-shape, -motility, and -adhesion [61]. Incorporation of exogenous VIM into cells is initiated in a perinuclear region, progressing in a polarized fashion toward the cell surface [62]. VIM possesses an organizing competence in the perinuclear filamentous network of a cell, which makes it important for protection of nuclear structural integrity and prevention of nuclear damage. Hence, suggesting the mechanical support provided by intermediate filaments to protect the nucleus and herewith the genome during migration [63,64,65].

In addition to the abovementioned markers, a subset of the placental bovine myofibroblast also expresses nuclear PGR, as demonstrated in the present study. In general, P4 is responsible for endometrial differentiation, secretory activity, myometrial quiescence, closure of the cervix and local immunotolerance [66,67,68,69]. In these regards, the PGR expression pattern may give information about myofibroblast responsiveness in the maternal compartment of the bovine placenta. Investigating these functional pathways in cattle placenta is certainly worth further attention as it could reveal important mechanisms underlying the regulation of bovine feto-maternal communication and subsequent detachment of fetal membranes from the maternal compartment. Furthermore, based on these results, the origin and functionality of fibroblast to myofibroblast differentiation in bovine placental tissue in comparison to other species should be the focus of future studies. E.g., decidualization in invasive placentae can be seen as another type of differentiation of fibroblasts, reflecting some similarities between these two processes, but there are still considerable differences [50,53,70,71].

On an ultrastructural level, it is highly likely that bovine placental myofibroblasts are related to fibroblastic cells, found in other tissues and organs [10,11,70]. However, there are no data available concerning the 3D ultrastructure and cell composition of bovine placenta myofibroblasts, which could help to better understand cellular compositions and mechanisms on the molecular biological level. Consequently, we performed a whole 3D cell reconstruction of the myofibroblast. The latter revealed an elongated cell type with various processes, which reflects their adjacent character enabling cell-to-cell-interactions and forming an interactive meshwork. In addition to that, stress fiber formation was observed, according to immunofluorescent and ultrastructural TEM findings.

Since the placenta represents a non-innervated organ, it has been postulated that placental cell function would be related to signal transduction mechanisms involved in the regulation of the blood flow in the fetal and maternal vessels, as well as in the possible shortening and lengthening of the chorionic villi providing adequate rhythmic for the feto-maternal metabolic exchange [72]. A similar mechanism is suggested for signal transduction. Nevertheless, even though myofibroblasts represent one of the main cell types in the maternal placenta stroma, positive for *α*-SMA, a deeper understanding on their contractability and control mechanisms is still missing. According to previously published studies on fibroblastic and myofibroblastic contractility in human dermal tissue, cellular contraction is inducible under in vitro conditions, without significant difference of generated forces between fibroblasts and myofibroblast [73].

Considering functionality in bovine maternal and fetal placenta units, the expression of angiotensin-II (Ang-II), as the main effector of the renin-angiotensin system [25,26], and prostaglandins (PGs) have been demonstrated in the bovine uterus and placental tissue [28,50]. PGs are also suggested to mediate placental development and fetal membrane detachment after parturition [74] and they may exert distinctive effects, e.g., contractive or relaxant, on smooth muscles in different organ systems [75]. Given their well-documented effects on smooth muscle contraction and extensive involvement in many smooth muscle-mediated physiological functions, PGs appear to be one of the most important epithelium-derived factors, that influence not only luteolysis, but also uterine contractility in different species [75,76]. Supporting myometrial contractility associated with fetal expulsion, PGs also have a vital role in the uterine activity during estrous cycle and pregnancy in the cow. This is represented in the expression of PGE2- and PGF2_*α*_-receptors, as well as former cyclooxygenase-1 and -2 (COX-1 and -2), currently known as prostaglandin-endoperoxide synthase-1 and -2 (PTGS-1 and -2), in a temporal and tissue-specific manner in bovine endometrial tissue, uterus, fetal membranes, and placenta, especially at the feto-maternal interface [30,31,32]. In detail, a high expression of PGF2*α* receptor (*FP,* current *PTGFR*) mRNA in caruncular tissue, showing no difference in tissue expression throughout gestation, was observed [32].

The PTGS-2 protein has also been investigated and revealed moderate and diffuse expression in epithelial cells of caruncular crypts and stromal cells of caruncular septum [32]. In mice, Ang-II promotes granulation tissue contraction and accelerates wound closure through one of its signaling receptors, angiotensin type 1 receptor (AGTR1), as well as the downstream TGF-β signaling pathway [27]. Furthermore, Ang-II binding sites are frequently observed in maternal fractions in the bovine placental tissue, equally expressed throughout the whole gestation period and essential for regulatory and growth processes also in the maternal stroma [25]. As previously described, mainly AGTR1 receptors but also low densities of AT2 receptors were observed in the mesenchymal tissue of the maternal part of the bovine placentome, especially close to the stalk and at the main branches of the maternal crypt [25]. Therefore, combined with the present CA findings, a AGTR1-effect in maternal placentomal unit is suggested. Summarizing the above-mentioned findings, PGF2_*α*_ and especially Ang-II are attractive candidates for inducing and stimulating myofibroblast contraction of placental tissue samples in vitro. Even though, PGF2_*α*_ needs further investigations since there was no potent contraction induction in the present placentomal caruncle tissue stimulation. Altogether, it seems reasonable to suggest that there may be a contractile effect which could be controlled by targeted drug stimulation. Nevertheless, further studies need to be conducted to gain a greater knowledge on these regulatory mechanisms.

Hence, in the present study, evaluation of in vitro contraction effects after tissue stimulation provided a more comprehensive characterization of the tissues’ contraction potential. In our investigation, Ang-II was a suitable in vitro contraction inducer, reflected as significant contractile effects on the tissue. These results highlight potential functional aspects of myofibroblasts in the bovine placenta, reacting to stimuli in a time-dependent manner. Even though the applicability of these conclusions is limited by the experimental models used in this study, i.e., the full section tissue slices. The manual separation of maternal (caruncular) tissue from the fetal cotyledon does not guarantee a full and clear separation of these tissues. After manual separation, cross-contamination of fetal and maternal cells can affect the specific assignment to the cell origin [77,78]. In the present study, this could lead to limitations in tracing the contraction effects to one cell type. Here we could demonstrate minimal fetal contamination of the separated caruncle tissue slices, confirming maternal cell origin and minimizing fetal cell effect. Nevertheless, it is suggested to conduct cell culture experiments on isolated maternal stromal cells to determine individual single cell contractive effects. Needless to say, considering the complexity of the contraction process and the lack of natural cell-to-cell interaction in an in vitro system, including different cellular compartments, long-term future outcomes obtained from cell culture studies need to be verified in vivo.

Mechanisms controlling parturition, placental ECM tissue remodelling as well as placental release in cattle are important to understand the timely expulsion of fetal membranes and its requirements, which might be either hormonally or mechanically driven [34]. Nevertheless, mechanisms responsible for pathological placenta attachment and detachment remain poorly understood. New insights on this field lead to closer understanding of pathogenic mechanisms and possible reasons of retained fetal membranes (RFM), a pathological condition with a high economic impact [34,79,80]. Deeper insights on endocrine control of parturition and placental tissue remodeling showed that e.g., P4 withdrawal only induces a limited spectrum of the processes related to normal parturition in late pregnant cows. Following this, P4 appears to be not crucial for the prepartal tissue remodelling in placentomes and the timely release of the placenta [81]. Performing gene expression profiling and investigating the molecular basis on bovine peripartal placentomes, findings of a previous study revealed three different physiological processes which play a fundamental role in placental detachment, i.e., apoptosis, degradation of extra cellular matrix and innate immune response [82]. However, there are no data available concerning the contractile impact and possible influence of myofibroblasts on the ECM remodeling in the bovine placental tissue during separation of placental compartments or pathological RFM. Results of this present study may give a first insight into the functional mechanisms and contractile activity of myofibroblasts in the maternal stroma.

## 5. Conclusions

In summary, we show ultrastructural features of myofibroblasts in the maternal stroma of the bovine placenta. The co-localization and expression of *α* -SMA, VIM and PGR was observed in caruncle stroma of pregnant and non-pregnant cows. Positive signals for Cx43 and FN suggest that myofibroblasts form interacting cell-networks and might be involved in ECM remodeling. TEM analyses and 3D reconstruction of the placental myofibroblasts demonstrate the ultrastructural composition of this cell type, reflected as stress fiber formation in the cell periphery, expanded cytoplasmic ER and perinuclear intermediate filament occurrence. More knowledge about stromal cell functionality was added with the contraction assay experiments, showing a significant contractile effect with Ang-II stimulation, but apparently not with PGF2_*α*_ treatment. According to the contractile potential of bovine placental stroma myofibroblasts, together with all presented results, we conclude that the functional impact of myofibroblasts can mainly be seen in wound healing, implying an important effect on post-partum involution processes of the uterus in cattle. Finally, the present work provides an ultrastructural basis for future investigations regarding the functional involvement of myofibroblasts around parturition and in placental separation process around term in the cow.

## Figures and Tables

**Figure 1 vetsci-10-00044-f001:**
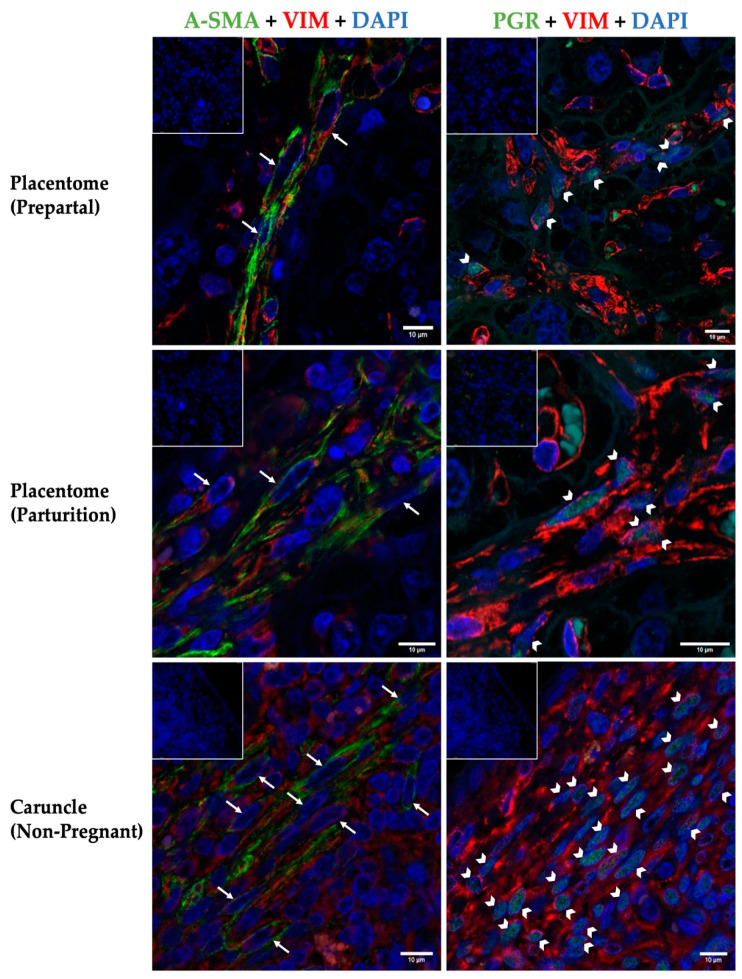
Three-color immunofluorescence staining of placental and NP caruncular tissue sections. Placentomes (prepartal and at parturition) from healthy cows and caruncle tissue samples from NP individuals were labeled with *α*-SMA (green, cytoplasm) or PGR (green, nucleus) and VIM (red). Cells which were positive for *α*-SMA and VIM (*α*-SMA+; VIM+) were identified as myofibroblasts in the left panel (arrows). In the right panel, myofibroblasts are reflected as PGR-positive cells (arrowheads) with additional VIM signal. Nuclei were additionally stained with DAPI (blue). In merged images nuclear signals for PGR and DAPI overlap and appear in brighter white/turquoise color. Autofluorescent signal was occasionally observed in erythrocytes. For each tissue section there was no background staining, shown in the negative isotype controls (insets).

**Figure 2 vetsci-10-00044-f002:**
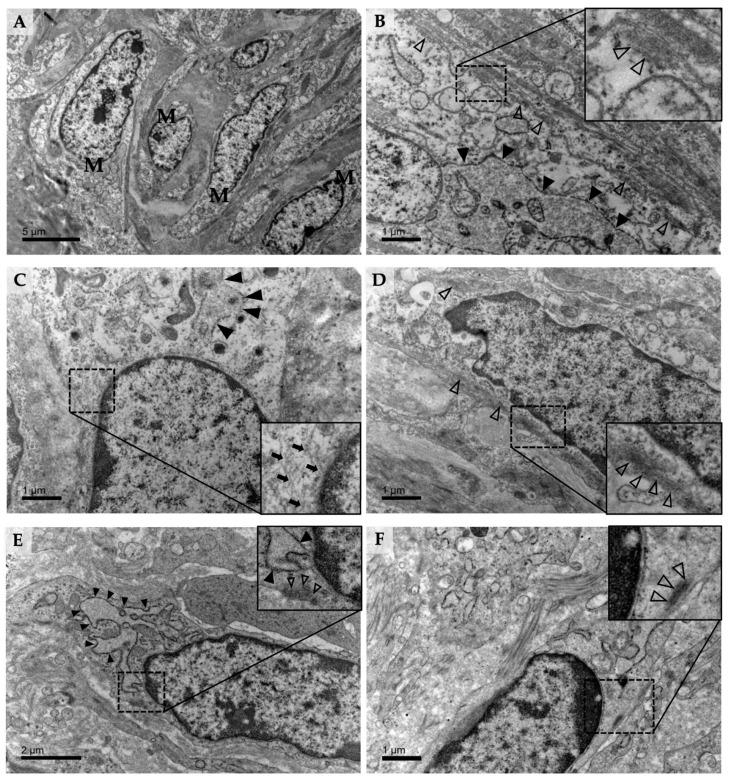
Transmission electron micrographs of bovine placentomes around parturition (**A**–**D**) or non-pregnant caruncle samples (**E**,**F**). (**A**) Ultrastructural overview of neighboring myofibroblasts (M). (**B**) Close up of a single myofibroblast, showing characteristic expanded cisternae of ER (filled arrowheads) and stress fibers (empty arrowheads) associated with either dense bodies or focal adhesions (inset). (**C**) Extended ER cisternae (full arrowheads) and perinuclear intermediate filaments (small arrows) are frequently observed. (**D**) A myofibroblast shows several strands and bundles of stress fibers (empty arrowheads), especially close to the cell membrane (inset). (**E**) Myofibroblast in the caruncular stroma. Representative enlarged ER cisternae (full arrowheads) and compacted stress fibers (empty arrowheads), attached to focal adhesions, can be detected (inset). (**F**) A single myofibroblast is demonstrated with characteristic stress fibers, attached to focal adhesions (dotted rectangle). A higher magnification is presented in the inset and shows the stress fibers attached to a focal adhesion at the cell membrane (empty arrowheads).

**Figure 3 vetsci-10-00044-f003:**
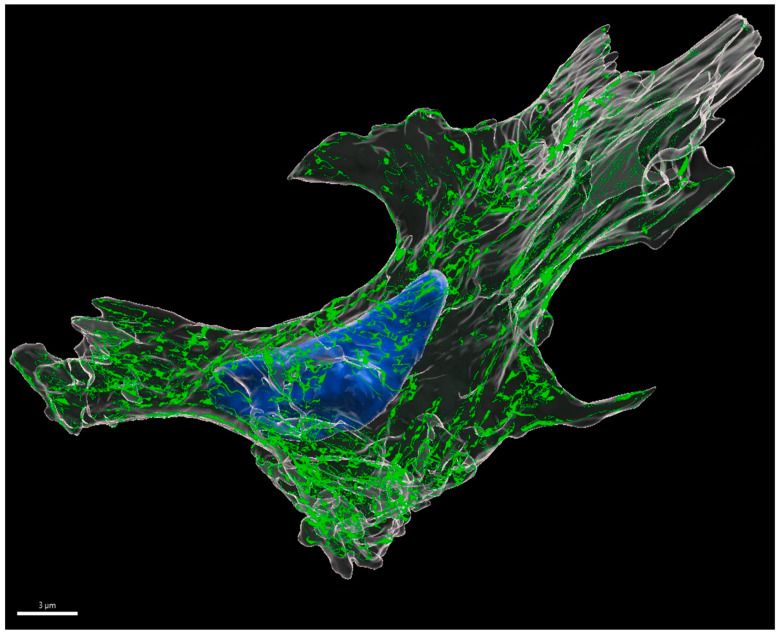
Presentation of a whole 3D-reconstructed myofibroblast in the maternal stroma of a pregnant cow (278 gd). The cell demonstrates processes in varying length and size, stress fibers (green) located in the cell periphery and the nucleus (blue), surrounded by the cytoplasm and cell membrane (white). Scale bar = 3 μm.

**Figure 4 vetsci-10-00044-f004:**
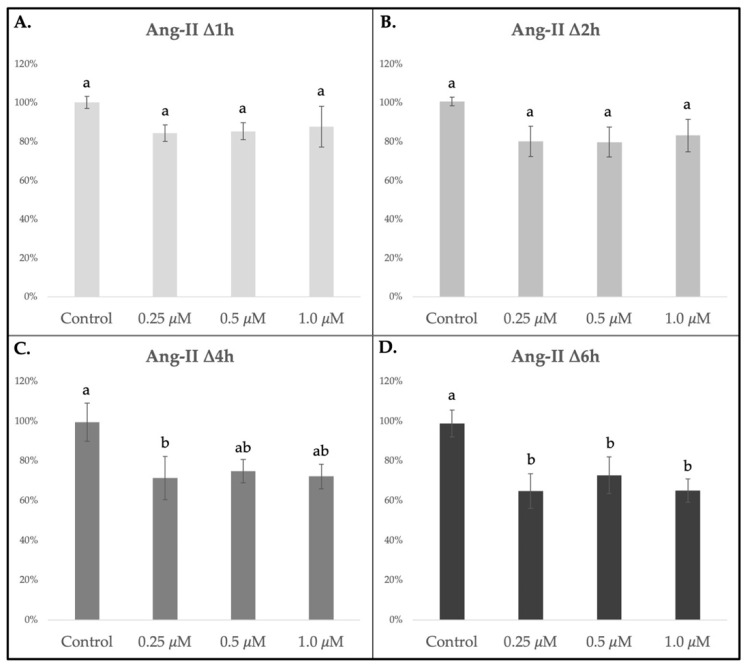
Arithmetic mean (±SD) area changes (in %) in Ang-II stimulated, placentomal caruncle tissue slices through contraction assay experiments. Slices were analyzed after different incubation times. (**A**) Area changes of Ang-II treated tissue slices after 1 h of stimulation (∆1 h). (**B**) Ang-II stimulated tissue slices after 2 h of incubation (∆2 h). (**C**) Area size changes of treated tissue slices after 4 h of Ang-II treatment (∆4 h). (**D**) Arithmetic mean area changes of Ang-II stimulated tissue slices after 6 h of treatment (∆6 h). Measurements started after preincubation and before the first treatment, which is considered as timepoint 0. To evaluate the contraction effect after in vitro stimulation, one-way ANOVA was applied, revealing: *p* = 0.03 for 4 h stimulation (**C**); *p* = 0.008 = highly significant for 6 h of incubation (**D**) When *p* < 0.05, analysis was followed by a Tukey–Kramer multiple comparison post-test. Lower-case letters (a,b) indicate significant differences (*p* < 0.05) among the concentration groups in the Ang-II treated slices during different incubation times.

**Table 1 vetsci-10-00044-t001:** Information on the tissue samples used in this study. * For each method, IF and TEM: n = 3.

Methods	Samples	Grouping and Number of Animals (n)	Reference or Collection Data
**Immunofluorescence (IF) staining**	Placentomes	**Pregnant (P)** Pre-partum n = 3 Parturition n = 3	Schuler et al. (2006)
**IF staining and Transmission Electron Microscopy (TEM)**	Caruncles	**Non-Pregnant (NP)** Post-partum n = 3 * Nulliparous n = 3 * Multiparous n = 3 *	collected at local slaughterhouse in Zurich
**TEM**	Placentomes	**Pregnant** Parturition n = 3	Boos et al. (2003)
**Contraction Assay (CA)**	Placentomal Caruncles	**Pregnant** n = 3	collected at local slaughterhouse in Zurich
**3D Reconstruction**	Placentome SBF-SEM Stack	**Pregnant** n = 1	Tiedje et al. (2021)

**Table 2 vetsci-10-00044-t002:** List of antibodies used for immunofluorescence (IF) staining.

Antibody	Company	Reference Number	Host	Dilution	Targets & Purpose
*α*-Smooth Muscle-Actin	Dako	GA611	Mouse monoclonal, anti-human	1:200	Myofibroblasts, primary antibody
Progesterone Receptor	Invitrogen	MA1-411	Mouse monoclonal	1:200	Nuclear expression in maternal stromal cells, primary antibody
Vimentin	Abcam	ab92547	Rabbit monoclonal	1:500	Mesenchymal cells, intermediate filament, primary antibody
Connexin 43	Abcam	ab11370	Rabbit polyclonal	1:200	Intercellular/gap junction marker, primary antibody
Fibronectin	Novus Biologicals	NBP1-91258	Rabbit polyclonal	1:200	Extracellular-Matrix (ECM), primary antibody
anti-Mouse IgG (H+L), Alexa Fluor Plus 488	Thermo Fisher	A32723	Goat polyclonal	1:200	Secondary Antibody
anti-Rabbit IgG (H+L), Alexa Fluor 594	Thermo Fisher	A-11012	Goat polyclonal	1:500	Secondary Antibody
Mouse IgG2a antibody	Exbio	11-458-C025	Mouse	1:200	Isotype Control
Mouse IgG antibody	Vector Laboratories	I-2000-1	Mouse	1:200	Isotype Control
Rabbit IgG antibody	Vector Laboratories	I-1000-5	Rabbit	1:400	Isotype Control

## Data Availability

The data presented in this study are available on a reasonable request from the corresponding author.

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
