# Peer review of "Ultrastructural and Immunohistochemical Characterization of Maternal Myofibroblasts in the Bovine Placenta around Parturition"

_vetsci, 2023, doi:10.3390/vetsci10010044_

Round 1
Reviewer 1 Report
In general, the paper provided an adequate introduction that reflects the current literature. The methodologies used are adequately detailed in the materials and methods section. The literature revision treatment is adequate and enough detailed for this type of work. Also, the discussion provides an appropriate treatment of the subject.
However, it is necessary to make corrections in the language (spelling and grammar) and the items specified below.
Acronyms/abbreviations should be defined the first time they appear in each section. In this sense, on line 37 reference is made to “prostaglandin F2?”, and subsequently, on line 103 it is expressed as “PGF2?”. Likewise, in lines 108 and 109, the term “PGF2” is used. It would be important to know if all times reference is made to the same element and, if not, it should be properly clarified.
Likewise, “extracellular matrix (ECM)” appears correctly defined on line 119. However, previously it appears undefined on lines 74 and 79.
Reviewer 2 Report
General comments
This manuscript, titled “Ultrastructural and immunohistochemical characterization of maternal myofibroblasts in the bovine placenta around parturition” aims to unveil some of the ultrastructural and functional aspects of myofibroblasts in the developed bovine placental stroma, mainly of the myofibroblasts in the maternal placental stroma.
This was a very interesting article to read, that add important and sound knowledge to the understanding of the late pregnancy placenta physiology and ultrastructure in cows. I agree that the present work provides an ultrastructural basis for future investigations regarding the functional involvement of myofibroblasts around parturition and in placental separation process around term in the cow, that can be useful for future works and help in creating processes to minimize the great problem of placenta retention in cows.
Some of the paragraphs are quite long, which makes them more difficult to read. I would suggest dividing them, if acceptable to the authors. When reading a scientific text is important to maintain focus, which is harder to do when paragraphs are much longer than 16-18 lines, in my opinion.
The work described in the manuscript was technically sound and correctly executed. The findings claimed are supported by the data gathered and the statistical analysis was correct. There is no suspicion of plagiarism. The bibliographic revision was thorough and up to date.
Title
The title is correct and well describes the manuscript.
Abstract
The Abstract clearly present the main features of the manuscript.
Key Words:
For a more effective indexing of the article, the Keywords should not repeat words already used in the Title.
Introduction
The Introduction section is, in general, well written and structured.
Line 83-84: As both sentences are from the same bibliographical reference there is no need for the first [18], after “gels”.
Materials and methods
This section describes all the M&M in detail and in a manner that makes it easy to objectively access all that was performed by the authors.
Table 2: In the final artworks it is important that a Table is all in the same page, for a clear reading and interpretation.
Results
The Results section is clear, objective and well written. The figures are exceptional and add a great interest to the reading.
Line 341: Add a comma between “In the TEM sections” and “we identified myofibroblasts.”
Discussion
The Discussion section is well written and addresses the necessary comparations with other published material. It is sometimes hard to follow the text due to the long length of the paragraphs and density of the text.
References
The References section is extensive and up to date.
Tables and Figures
The Tables and Figures are correct and add to the interest of the manuscript.
